# Statistical Evaluation of Competing Models in Brain MRI Analysis

**Ekaterina Kondrateva**[1]                                          KONDRATEVA@GMAIL.COM
**Aleksandr Yugay**[1]                                    SASHA.YUGAY99@GMAIL.COM
[1] *Skolkovo University of Science and Technology*

## Abstract

Current state-of-the-art models in medical image analysis are often evaluated using small, single train/test split datasets, which may not provide a reliable indication of model improvement. This practice, combined with the increasing complexity and computational cost of newer architectures, exacerbates reproducibility issues, particularly in brain MRI imaging. This paper introduces a robust statistical framework to assess model comparisons in medical imaging. We correct common misapplications of the t-test, which assumes data normality, and promote the use of Two One-Sided Tests for demonstrating non-inferiority in model optimization scenarios. Through this approach, we aim to enhance the rigor of model performance evaluations and address the challenges of dataset size and test validity in medical image analysis.

**Keywords:** SOTA, classification, segmentation, result significance, benchmarking, evaluation

## 1. Introduction and Previous works

In medical deep learning and computer vision, it is common to encounter small and heterogeneous datasets while simultaneously deploying state-of-the-art (SOTA) models such as 3D U-net (Çiçek et al., 2016), UNetR (Hatamizadeh et al., 2022), or ensembles of 2D U-nets (Rad et al., 2018; Couteaux et al., 2019; Dang et al., 2024). Datasets with several hundreds of subjects, including UCLA LA5c Study (Gorgolewski et al., 2017), focus on classification tasks for conditions like schizophrenia and ADHD. In contrast, ABIDE (Di Martino and Mostofsky, 2023), ADNI (Petersen et al., 2010), and BraTS (Bakas et al., 2018), with thousands of subjects, handle tasks such as autism classification and brain tumor segmentation BraTS (Bakas et al., 2018). However, many papers at leading conferences still report results comparing to SOTA using a single train/test split. Here we advocate for a pivot in statistical methods commonly used in two-model comparisons. The most known test is the 't-test' (Kim, 2015), which often appears in papers simply marked with a 'p-value' star in the results table and is especially rarely exploited in classification papers.

## 2. Methods

We will explore two typical settings and assess three different statistical methods for two common tasks in data analysis: Classification and Semantic Segmentation.We consider two sets of hypotheses: **1.** Our model is not similar to the baseline; this is exactly what is tested with a two-sided t-test, which is commonly used in numerous works. this is the most

typical scenario when you want to demonstrate that your model is state-of-the-art (SOTA). **2.** Our model is not worse than the baseline model. This is another common scenario, or the difference between the models is not more than $\Delta$. This scenario is used when optimizing the model or conducting distillation, or checking if your model is not worse than SOTA.

**Statistical Tests: 1.** The t-test, applicable for both paired and independent samples (Kim, 2015), and its counterpart, the Welch test (Welch, 1947), which operates without the assumption of equal variances. **2.** The Wilcoxon signed-rank test for independent samples (Wilcoxon, 1992). **3.** Two one-sided tests (TOST) for equivalence (Schuirmann, 1987), which are particularly useful for proving non-inferiority or equivalence between two competing models.

**Multiple Comparison Correction**. Given that the comparisons we perform are part of a set of experiments, we apply the Bonferroni correction (Weisstein, 2004) to adjust the p-values, ensuring the control of the family-wise error rate.

## 3. Results

**1. Classification** In this study, we compare two models for ADHD classification using data from the LA5 study with 186 subjects. We employ ROC-AUC as the evaluation metric in a 3-fold cross-validation setup with a fixed random seed. The reported mean ROC-AUC values for the "baseline" and "ours" models across three folds are 0.630 (0.049) and 0.716 (0.065), respectively. First, we assume one fold from CV as a single observation, thus we have 3 samples (Table 1, rows 1-8). Secondly, we use per-subject prediction accuracy as $|y - y_{\text{pred\_proba}}|$, where $y$ represents the true labels, and $y_{\text{pred\_proba}}$ the prediction probability of the classifier. To determine the significance of the observed improvement in predictive accuracy of the proposed model over the baseline, we test four model architectures and configurations. To maintain an overall significance level of 0.05, we adjust the individual hypothesis testing threshold to $0.05/4 = 0.0125$ using the Bonferroni correction. Our analysis assesses the null hypothesis (H0) that there is no significant difference in mean performance between the two models, contrasting it with the alternative hypothesis (H1) proposing a significant difference, either in both directions ('two-sided') or exclusively in one direction ('greater'). In Table 1, we show that H0 is rejected only in the last test.

**2. Segmentation** Our goal is to statistically confirm that our model performs similarly to the baseline but with significantly faster inference. We analyse TCIA-GBM (Clark et al., 2013) with 102 subjects with brain tumors. We use Dice as the evaluation metric in a 3-fold CV setup with a fixed random seed. The reported mean Dice values for the "baseline" and "ours" models across three folds are 0.845 (0.112) and 0.849 (0.116), respectively.

We use Wilcoxon test, similar to previous tests, to assess the null hypothesis (H0) that there is no significant difference in mean performance between the two models. This is contrasted with the alternative hypothesis (H1), which proposes a significant difference in one direction ('greater').For the Two One-Sided Tests (TOST), the null hypothesis (H0) posits that the two samples are significantly different. The alternative hypothesis (H1), meanwhile, suggests that the means are not significantly different within a specified equivalence margin. We set the margin $\Delta$ based on the benchmark performance of the nn-U-Net model on the BraTS dataset from the original study (Isensee et al., 2021), specifically for the tumor core: Mean (STD) 0.851 (0.024), taking $STD/2 = 0.012$.

Table 1: H0 testing for two classification models comparison (pvalue<0.0125). For related samples cross-validation split should be fixed. * - data normality and ** - normality and variance equality should be proven.

|    | Statistical Test | Samples | Observations | Alternative | pvalue | H0 |
|----|------------------|---------|--------------|-------------|--------|-----|
| 1  | Wilcoxon | Related | 3 | Two-sided | 0.5 | Accept |
| 2  | Wilcoxon | Related | 3 | Greater | 0.25 | Accept |
| 3  | T-test** | Independent | 3 | Two-sided | 0.245 | Accept |
| 4  | Welch test* | Independent | 3 | Two-sided | 0.144 | Accept |
| 5  | T-test, Mean(STD)* | Related | 3 | Two-sided | 0.139 | Accept |
| 6  | T-test** | Independent | 3 | Greater | 0.122 | Accept |
| 7  | Welch test* | Independent | 3 | Greater | 0.072 | Accept |
| 8  | T-test, Mean(STD)* | Related | 3 | Greater | 0.069 | Accept |
| 9  | Wilcoxon | Related | 186 | Two-sided | 0.013 | Accept |
| 10 | Wilcoxon | Related | 186 | Greater | 0.007 | **Reject** |

Table 2 shows that the first two tests accept H0, which does not prove equivalence between the two models. However, the third experiment demonstrates that the two models are equivalent within the chosen margin.

Table 2: H0 testing for comparison of two segmentation models with Dice scores per-subject (on comparison in set of experiments). TOST $\Delta = 0.012$ (Dice). After Bonferroni correction, desired pvalue $< 0.0125$. * - data normality should be proven.

|   | Statistical Test | Samples | Observations | Alternative | pvalue | H0 |
|---|------------------|---------|--------------|-------------|--------|-----|
| 1 | Wilcoxon | Related | 102 | Two-sided | 0.029 | Accept |
| 2 | Wilcoxon | Related | 102 | Greater | 0.014 | Accept |
| 3 | TOST* | Related | 102 | Equivalent | 0.007 | **Reject** |

## 4. Conclusions and Discussion

**1.** We recommend using the Wilcoxon test for related samples with a 'greater' alternative to demonstrate the significance of differences. For model classification, we propose assessing per-subject accuracy.
**2.** When using T-test and TOST methods, the assumption of data normality should be verified (Razali et al., 2011; Shaphiro and Wilk, 1965).
**3.** For assessing the equality of model results, we suggest employing two one-sided tests (TOST) analysis.
**Assumptions:** Most of the statistical tests, both parametric and non-parametric, that we have mentioned operate under the "big numbers" rule.

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
