# OpenReview forum: "Statistical Evaluation of Competing Models in Brain MRI Analysis"
_MIDL.io/2024/Short_Papers — MIDL 2024 Short Papers_

### Official Review · Reviewer_VPvr · 2024-04-18

**Confidence:** 4
**Final Rating:** 5

**Review:**

The paper is a kind of tutorial on statistical significance evaluation of models being better or not worse than one other, illustrated with a classification and segmentation problems in brain MRI.

I think this is much needed work because the level of evaluation of some papers in our field is quite lacking. I am not 100% sure the proposed methodology solves all the problems – in particular, treating cross-validation folds as independent samples seems questionable – but taking steps such as these would already go a long way.

Given the importance of the topic, as well as originality (something different than yet another model architecture) and clarity of presentation, I would highly recommend presenting/discussing this work at the conference.

---

### Decision · Program_Chairs · 2024-04-26

Accept